# Mechanisms of multi-species mealybug invasions in Hainan Island of China: Integrating niche, distribution, and habitat drivers

Qin Si[1,2☯], Jie Hu[3☯], Zhong Hua[4], Jian Wang[3], Mulan Ji[3], Gaochao Xu[1], Binxin Wu[1,2,5*], Yanjing Zhang[3*]

**1** Jiangsu Maritime Institute, Nanjing, Jiangsu, China, **2** College of Biosystems Engineering and Food Science, Zhejiang University, Hangzhou, Zhejiang, China, **3** Nanjing Institute of Environmental Sciences, Ministry of Ecology and Environment, Nanjing, Jiangsu, China, **4** Zhejiang Business Technology Institute, Ningbo, Zhejiang, China, **5** College of Chemistry and Chemical Engineering Jinggangshan University, Ji'an, Jiangxi, China

☯ These authors contributed equally to this work.
* yanjingzhang6688@163.com (YZ); bwu66@zju.edu.cn (BW)

## Abstract

Mealybugs, highly invasive pests causing global agricultural damage, threaten China's tropical Hainan Island—a critical biosecurity zone. This study investigates spatial patterns, interspecific interactions, and environmental drivers of 15 invasive mealybug species using integrated ecological niche modeling (Maxent), niche/range overlap analyses, Joint Species Distribution Models (JSDM), and Structural Equation Models (SEM). Our research revealed that a strong coastal-inland richness gradient emerged, where humid tropical climates and monoculture plantations supported 11–15 species in eastern coasts, whereas topographic complexity limited invasions to 0–5 species in forested mountains. In parallel, high niche overlap ($I \geq 0.67$) among species reflected climate-mediated thermal tolerance and habitat-driven broad host preferences. Range overlap patterns diverged: high-overlap pairs (e.g., *Planococcus minor* and *Paracoccus marginatus*, $I > 0.8$) coexisted via adaptation to 24–28°C and generalist hosts, whereas low-overlap pairs (e.g., *Icerya purchasi* and *Phenacoccus nigra*, $I = 0.37$) segregated through host specificity. Mechanistically, JSDM confirmed these patterns, revealing competitive exclusion where resource overlap and thermal adaptation divergence occurred (e.g., *P. marginatus* vs. *Dysmicoccus brevipes*), versus coexistence promoted by host specialization (e.g., *Ceroplastes psidii*'s rubber tree specificity versus *P. marginatus*'s generalism), reproductive strategy divergence (parthenogenesis vs. seasonal outbreaks), and external environmental factors. Ultimately, SEM analyses and Linear regression identified significant positive correlations between habitat conditions and species suitability ($R^2 = 0.71$, $p < 0.001$), with habitat type as the dominant driver (total effect = 0.48), where climate and topography indirectly regulated suitability through habitat characteristics (e.g., elevation, latitude)

**Data availability statement:** All relevant data are within the manuscript and its Supporting Information files.

**Funding:** The author(s) received financial support for this work from the Intergovernmental International Science and Technology Innovation Cooperation Program under the National Key Research and Development Plan (Grant No. 2024YFE0198600 to YZ) and the Science and Technology Innovation Fund of Jiangsu Maritime Institute (Grant No. 2023BSKY09 to QS).

**Competing interests:** he authors have declared that no competing interests exist.

while pest-infested areas directly enhanced suitability (path coefficient = 0.24). Our research framework elucidates multi-species invasion assembly mechanisms, new insights and theoretical support for the management of 15 invasive mealybug species, providing a solid foundation for optimizing future species distribution models, and validating the methodological value of integrated modeling (JSDM-SEM) in disentangling invasion complexities.

## Introduction

Biological invasions are widely recognized as major threats to biodiversity, ecosystem services, and socioeconomic stability, as emphasized by the Kunming-Montreal Global Biodiversity Framework under the Convention on Biological Diversity (CBD) [1–2]. As a classic example of a biological invasion, invasive pests disrupt natural habitats and damage agricultural productivity, leading to substantial economic losses [3–4]. Mealybugs (Hemiptera: Coccoidea: Pseudococcidae) are small insects covered with white or creamy-yellow waxy secretions, often described as "dust-like" in appearance. Over 2000 species have been documented globally (more than 1,000 in China), many of which are significant agricultural and forestry pests in tropical and subtropical regions, while some frequently infest greenhouse-cultivated plants in temperate zones [5–8]. This group exhibits typical invasive traits, including small body size, cryptic behavior, morphological homogeneity, broad host ranges, rapid reproduction, and high adaptability [9]. Their spread via international trade of fruits, vegetables, and transport vehicles enables colonization, range expansion, and outbreaks, posing significant control challenges [10]. Mealybugs infest a wide variety of hosts, such as fruit trees, crops, and ornamental plants [11–12]. Our field surveys in Hainan confirm their polyphagous impacts on the island, primarily damaging cash crops (e.g., papaya, cassava, rubber), fruit trees (e.g., guava, coconut), and ornamental plants (e.g., Hibiscus rosa-sinensis), with secondary infestations in vegetables (e.g., eggplant, pepper). Adults and nymphs of mealybugs cluster on branches, leaves, and fruits to feed on sap, while their honeydew excretion induces sooty mold (*Capnodium* spp.), severely weakening plant vigor and leading to gradual wilting and death of affected individuals [7]. In extreme cases, infestations can cause complete crop failure or render agricultural products economically worthless [5,12]. Invasive mealybugs often form obligate mutualisms with native or co-invasive ants, exacerbating ecological impacts through resource-mediated synergies. For instance, the invasive red imported fire ant (*Solenopsis invicta*) boosts honeydew production by sheltering mealybugs, creating a positive feedback loop that amplifies population outbreaks [13–15].

The omnivorous nature and diverse parasitic habits of mealybugs, coupled with accelerated economic globalization and agricultural trade liberalization, have triggered a global surge in alien mealybug invasions [10,16]. In China, invasive species—including *Phenacoccus solenopsis* and *Dysmicoccus neobrevipes*—have caused severe economic losses through direct crop damage, secondary pathogens,

and trade restrictions [17–18]. A notable case is *D. nebrevipes*, which destroyed over 100,000 acres of sisal plantations in Guangdong after its 1998 Hainan introduction [12]. Hainan Island, as China's free trade port, faces heightened risks: field surveys reveal infestations across 4,880 km², where weak native resistance and year-round warm humidity accelerate invasions [19]. This climate-mediated vulnerability, compounded by international trade networks, creates synergistic threats to both ecosystems and regional economies.

Understanding the potential habitat suitability of mealybugs is a cornerstone of effective pest management, as their spatial dynamics are profoundly influenced by abiotic and biotic drivers. Climatic variables—particularly temperature and precipitation regimes—dictate species' physiological tolerances and dispersal corridors, with global warming amplifying range shifts and phenological mismatches between pests and host plants [20–22]. Beyond abiotic constraints, biotic interactions (e.g., competition, mutualism, and predation) critically mediate population persistence and range boundaries. Although historical models largely neglected these interactions beyond local scales, recent syntheses reveal their pervasive influence across spatial extents by altering species' climatic niches and assembly rules [23]. For instance, mutualistic associations with ants can override climatic limitations, enabling invasive mealybugs to colonize suboptimal habitats [13]. However, current studies of species habitat suitability often fail to clarify the relative contributions of environmental filtering and species interactions, especially for trophically complex taxa like mealybugs.

The Maximum Entropy (MaxEnt) model, recognized for its precision, operational stability, and efficiency, exhibits higher predictive capacity than other models while demonstrating notable resilience to sampling bias. It has been widely applied to forecast suitable habitats for invasive pests, including *P. marginatus* [24], *S. invicta* [25], and *Wasmannia auropunctata* [26]. ENMTools is a set of comparative similarity measures and statistical tests that allow quantitative comparison of ecological niche models. It can be used to measure niche and range overlap among species distributions [27–28]. Joint Species Distribution Models (JSDMs) decompose species co-occurrence patterns to quantify environmental filtering and biotic interactions [29–30], outperforming species distribution models (SDMs) in reconstructing environmentally driven distributions under data-limited scenarios. Critically, JSDMs enable simultaneous multi-species modeling by leveraging occurrence data from well-sampled species to inform predictions for data-deficient taxa [30].The Hierarchical Modelling of Species Communities (HMSC) is a Bayesian extension of JSDMs. Structural Equation Modeling (SEM) surpasses traditional multivariate methods by detecting causal networks to resolve synergistic relationships in complex ecosystems [31–32]. Collectively, these approaches enhance our capacity to decipher interactions among environmental drivers, species relationships, and habitat transformations.

The fragile island ecosystem of Hainan and the extraordinary invasiveness of mealybugs (Pseudococcidae) together constitute a unique biological challenge, posing severe economic threats to agriculture and forestry. This urgently demands a multidimensional analysis of invasion drivers to achieve precise prevention and control. This study focuses on Coccoidea, a globally significant invasive agricultural pest, using Hainan Province - China's only tropical island region – as the research area. We innovatively explored multidimensional analytical approaches including Maxent-based ecological niche modeling, ENMTools-based niche and range overlap analysis, HMSC, and SEM implemented through the *lavaan* R package to systematically investigate the complex invasion mechanisms and multi-scale driving factors of 15 high-risk invasive mealybug species. Our research will comprehensively analyze the niche differentiation characteristics, interspecific interaction networks, and environmental driving mechanisms underlying multi-species mealybug invasions in tropical island ecosystems.

Methodologically, this study advances through four key dimensions: prediction of suitable habitats for 15 invasive mealybug species using Maxent with subsequent habitat overlap analysis to assess species richness, habitat niche and range overlap analysis via ENMtools, construction of interspecific interaction heatmaps through HMSC and JSDM to reveal competition and synergy patterns, and application of SEM with *lavaan* to quantify environmental drivers of habitat suitability. The findings provide crucial scientific evidence for advancing invasion ecology theory while offering methodological support for mealybug management in tropical agroecosystems, demonstrating dual theoretical and practical value for addressing biological invasions under climate change.

 

## Materials and methods

### Multi-step habitat suitability models

The study investigated 15 invasive mealybug (Coccoidea) in Hainan Province, China, each assigned a letter code for reference: a) *Dysmicoccus brevipes* (pineapple mealybug); b) *Icerya purchasi* (cottony cushion scale); c) *Planococcus minor* (Pacific mealybug); d) *P. solenopsis* (solenopsis mealybug); e) *Pseudococcus jackbeardsleyi* (Jack Beardsley mealybug); f) *Phenacoccus madeirensis* (Madeira mealybug); g) *Phenacoccus marginatus* (papaya mealybug); h) *P.manihoti* (cassava mealybug); i) *Phenacoccus parvus* (lantana mealybug); j) *Phenacoccus lilacinus* (lilac mealybug); k) *Aulacaspis yasumatsui* (cycad aulacaspis scale); l) *Ferrisia virgata* (striped mealybug); m) *Ceroplastes rusci* (fig wax scale); n) *D. neobrevipes* (grey pineapple mealybug); and o) *Parasaissetia nigra* (nigra scale).

All occurrence data (1,680 raw records) were obtained through systematic transect-based surveys conducted across all 18 land-based cities/counties of Hainan Island from 2023 to 2024. The surveys spanned coastal agricultural zones (e.g., Wenchang, Sanya) to inland mountainous areas (e.g., Wuzhishan, Qiongzhong), documenting GPS coordinates, elevation, occurrence habitats, and damage areas of mealybug infestations, with emphasis on vulnerable agricultural and forestry regions. To mitigate spatial autocorrelation-induced model bias [33], we implemented spatial thinning using ENMTools [34]: A single record was retained per 1 × 1 km grid cell, yielding 855 spatially independent occurrences for Maxent. This preprocessing step effectively reduced sampling bias from clustered observations, ensuring the models reflect authentic species-environment relationships rather than artificial aggregation patterns.

Environmental variables were obtained from the WorldClim database (version 2.1), comprising 19 bioclimatic variables representing key climatic dimensions including temperature, precipitation, and humidity. These variables, derived from global climate surfaces interpolated at 30-arcsecond resolution (~1 km²), were cropped to Hainan Island's geographic boundaries and resampled to 1-km spatial resolution to match species occurrence data. Prior to modeling, multicollinear variables were removed through sequential Spearman correlation ($|r| \geq 0.8$) and variance inflation factor (VIF ≥ 10) thresholds [35–36].

The Maxent (version 3.4.4) was employed to predict current suitable areas for invasive species using occurrence records and environmental variables. Model performance was evaluated using the minimum information criterion (Delta AICc) to balance fit and complexity [37–38], and model predictions were evaluated using AUC values [39]. Execution parameters included: 25% of data randomly allocated as test set (75% training), 10,000 background points, 5,000 maximum iterations, and 10 replicate runs (Subsampling method). The model configuration was set to Cloglog output format [40], with jackknife validation, environmental response curves, and ASC file format enabled. To further validate model robustness, 10 replicated runs were performed using optimized parameters, while retaining default values for other parameters. Species richness (the number of different species per grid cell) was calculated by superposing all of the species ranges. Finally, the results of habitat suitability of 15 species of mealybugs in Hainan were shown by the richness map of the suitable habitat area.

### Niche and range overlap

ENMTools is a set of comparative similarity measures and statistical tests that allow quantitative comparison of ecological niche models. It can be used to measure niche and range overlap among species distributions [27–28]. We used EnMtools v1.3 (http://purl.oclc.org/enmtools) to measure niche and range overlap among potential distributions of 15 invasive mealybug.

Niche overlap between species pairs was quantified using the *I* statistic derived from Hellinger distance [28,41], which measures similarity in environmental response profiles. Compared to conventional metrics (e.g., Schoener's D), the *I* index demonstrates reduced sensitivity to extreme environmental outliers while maintaining robust discriminatory capacity for detecting niche divergence [34]. Both niche overlap (*I*) and geographic range overlap were scaled from 0 (no overlap) to 1 (complete overlap), with higher values indicating greater ecological similarity and spatial co-occurrence, respectively.

Analyses were conducted in N-dimensional environmental space to avoid PCA-induced information loss [42]. ENMTools' integrated null hypothesis testing further distinguished biologically driven niche shifts from stochastic background variation. The unified interface enabled batch processing of Maxent-derived suitability rasters, ensuring computational reproducibility across species pairs (via lapply function).

## Species interaction with HMSC and JSDM

We implemented the HMSC to quantify interspecific interactions among 15 invasive mealybug species. The HMSC framework advances JSDM methodology by hierarchically integrating species traits, spatio-temporal structures, and phylogenetic relationships within a unified Bayesian framework [43–44], while employing latent variable (LV) modeling through linear regression to disentangle environmental covariate effects from residual co-occurrence signals driven by biotic interactions [45–46].

Firstly, the distribution data for the 15 invasive mealybug species were organized and converted into a binary matrix of 0 and 1, where "1" indicates the presence of the species in a grid cell and "0" indicates its absence. Simultaneously, environmental variable data for each grid cell were collected. Secondly, Within the HMSC framework, a separate distribution model was developed for each species, incorporating interspecific interaction terms. These interactions were modeled through synergistic and competitive effects, where positive values indicate synergism and negative values indicate competition [47,48]. The influence of environmental variables was incorporated through their direct effects on species distributions. Thirdly, the model was fitted using the Markov Chain Monte Carlo (MCMC) method to estimate model parameters and their uncertainties. Model fit was assessed by examining the convergence of the posterior distributions and effective sample size, and the Widely Applicable Information Criterion (WAIC) was used to select the best-fitting model [30]. By calculating the interaction effects between species, we analyzed their competitive and synergistic relationships. For each pair of species, the interaction coefficient was computed and inferred through posterior distributions. Finally, an interaction heatmap was generated to visualize the ecological relationships among the different species.

## Integrating multi-scale drivers of habitat suitability

This study employed SEM to investigate the drivers of habitat suitability for 15 invasive mealybug species in Hainan of China, focusing on direct and indirect effects of climatic factors (temperature, precipitation), topography (latitude, elevation), habitat type, and infestation area. The model was implemented using the *lavaan* package in R [31], leveraging SEM's capacity to analyze multivariate causal networks. Guided by a posteriori logic [49], we constructed a hypothesized network of variable relationships (causal or correlational) based on ecological expertise. Following model specification, we assessed overall fit indices and tested significance of individual pathways [50–51] to validate model utility. Total effects were calculated as the sum of direct and indirect effects between variables [49]. Results were visualized using a graphical framework: unidirectional arrows denote causal relationships (independent→response variables), while bidirectional arrows represent correlations.

The response variable, habitat suitability, was derived from Maxentpredictions. Explanatory variables included: 1), temperature and precipitation data extracted from WorldClim, synthesized into a single composite climatic factor via Principal Component Analysis (PCA); 2) habitat types and pest area from field surveys; and 3) latitude records from surveys and elevation data from WorldClim. To address scale differences and meet model assumptions, continuous variables (e.g., climatic composite, occurrence area, elevation) were standardized using Z-scores, while right-skewed variables (e.g., habitat types) underwent $log(x+1)$ transformation to improve normality [52]. Habitat types encompassed human-dominated landscapes (e.g., rural roads, orchards, village lands), agricultural production systems (irrigated/paddy fields), hydrological features (ditches, rivers, reservoirs), urban functional areas (transportation lands, park green spaces), and specialized land uses (rubber plantations, mining lands).

Guided by hierarchical niche theory [53], our framework conceptualizes habitat suitability determination as a tri-scale process: macro-scale climatic and geographic drivers (climatic gradients, latitude), meso-scale habitat mediators (habitat type diversity, infestation area's area), and micro-scale suitability responses. This hierarchy operationalizes the "climate-habitat-suitability" cascade hypothesis through four predefined pathways: 1) direct climatic effects on suitability via thermal tolerance thresholds; 2) indirect climate impacts mediated by habitat restructuring (e.g., temperature-driven shifts in forest-grassland mosaics); 3) geographic gradients (latitude, elevation) as spatial covariates to account for climate-habitat heterogeneity; and 4) habitat type's autonomous regulatory capacity, anticipated to demonstrate predominant direct effects based on prior ecological studies.

The structural equation model (SEM) was validated using established goodness-of-fit criteria: Comparative Fit Index (CFI) >0.90 (excellent fit: >0.95), Root Mean Square Error of Approximation (RMSEA) <0.05 (good fit), <0.08 (acceptable), Standardized Root Mean Square Residual (SRMR) <0.08, and Goodness-of-Fit Index (GFI) >0.90 (optimal: >0.95) [54–57]. Modification indices (MIs) were examined to identify potential model misspecifications. Final model adequacy was determined through holistic interpretation of all indices, ensuring rigorous evaluation of SEM performance. Additionally, we examined Modification Indices (MI) to identify potential missing paths. All indices were interpreted collectively to determine overall model adequacy, ensuring robust evaluation of the SEM's performance.

## Results and analysis

### Species richness in the current climate context

Our results illustrate the spatial distribution of the abundance of 15 species of mealybugs in Hainan Province (Fig 1). The species richness of the province was categorized into three different levels: 0–5 species (low), 6–10 species (medium) and 11–15 species (high) (Fig 1a). Areas with low species richness (shown in green), corresponding to 0–5

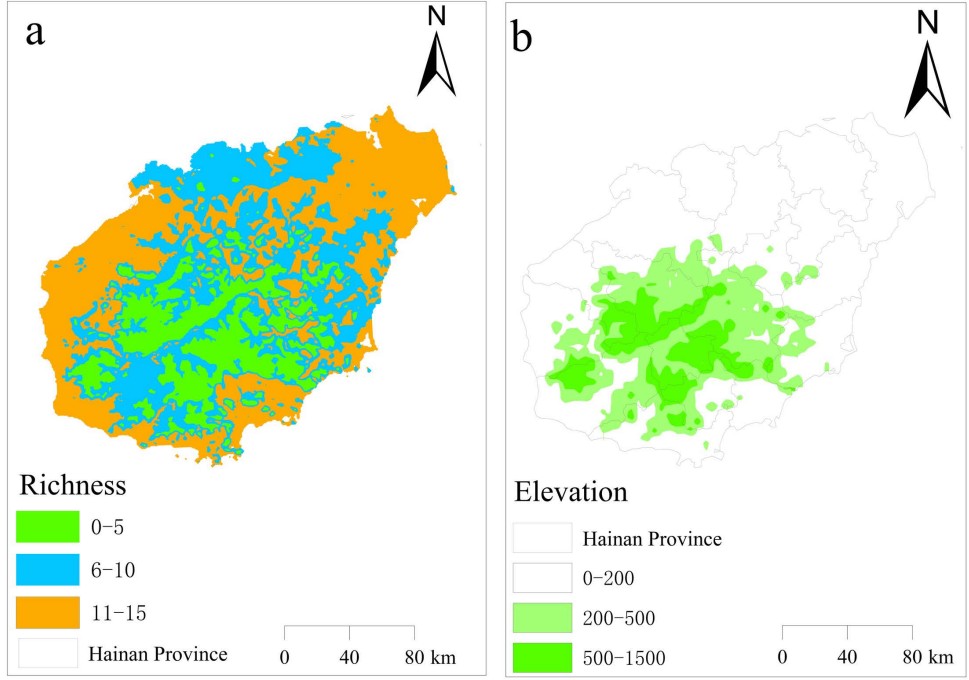

**Fig 1. Spatial Visualization Map of 15 invasive mealybug (Coccoidea) species richness (a) and topographic elevation (b) in Hainan Island.**

invasive species, cover 6,563.80 km² (18.54% of Hainan's total area). These regions are predominantly located in the central-southern and southwestern mountainous areas, particularly in Wuzhi Mountain and Yingge Ridge (Fig 1b).

Areas with moderate species richness (6–10 invasive mealybug species, depicted in blue) cover 13,213.32 km² (37.33% of Hainan's total area), primarily distributed in peripheral zones adjacent to low-richness areas and across central/eastern Hainan. Their relatively homogeneous spatial pattern suggests these ecosystems exhibit intermediate invasion resistance – stronger than low-richness zones but weaker than high-richness hotspots.

High-richness invasion hotspots (11–15 species, shown in orange) span 15,622.88 km² (44.13% of the province), predominantly clustered in northeastern and coastal regions. These areas demonstrate optimal invasion conditions resulting from synergistic factors: 1) coastal proximity facilitates exotic species introduction; 2) intensive human activities (tourism, international trade) enhance dispersal vectors; and 3) low-elevation topography coupled with simplified vegetation structure (dominated by ornamental species) creates homogenized habitats. Together, these factors establish ideal environments for invasive species establishment and expansion.

## Niche and range overlap analysis

Multidimensional environmental niche assessment using ENMTools (Hellinger's I index) revealed significant environmental niche convergence among the 15 invasive mealybug species (Fig 2). Niche overlap values ($I$) ranged from 0.67 to 0.98, with all species pairs except *I. purchasi* (b) and *P. nigra* (o) exhibiting high overlap ($I > 0.7$, Warren et al., 2008), indicating broad environmental adaptability convergence. Extreme convergence ($I \geq 0.9$) was observed in specific species pairs: species a with i/h/e; species c with o/k/i/m/l/h/g/f/e/d; species d with all others; species e with o/m/l/i/h/g; species f with m; species g with o/m/l/k/j/i/h; species h with o/m/l/i; species i with n/m/l/k; species j with n/m/l/k; species k with o/m/l; and species l with o/m.

Range overlap analysis using the Sørensen index (lower triangle in Fig 2) revealed a bimodal distribution of co-occurrence probabilities ranging from 0.37 to 0.87. High overlap pairs (Sørensen > 0.8) included species c with g/h/k/m/n, species d with g/k/l, species e with g/h/m, species g with l/m/o, and species l with m, suggesting extensive shared suitability under current conditions. However, non-overlapping areas persisted in high-overlap pairs (e.g., c-g), implying constraints from resource limitations or biotic interactions (e.g., competitive exclusion). The low niche–low distribution overlap pair (b-o, $I = 0.67$/Sørensen = 0.37) further highlighted the synergistic effect of environmental divergence and spatial isolation.

## Species interaction analysis

We used JSDM to analyze the interactions among 15 invasive mealybug species, presenting the strength and direction of these interactions through a heatmap. The results reveal pronounced spatial heterogeneity in species interactions (Fig 3). The figure displays the interspecies relationships, with the intensity of interactions indicated by color gradients: red areas represent positive synergistic effects, while blue areas denote negative competitive effects.

These invasive mealybug species displayed red-positive synergistic relationships, with stronger positive correlations predominantly observed among the following species pairs: Species b with d, i, n; Species c with e, h, m; Species d with b, i, k, n; Species e with c, g, o; Species f with j, l; Species g with e, l, o; Species i with b, d, n; Species j with f, n; Species k with d, o; Species m with c; Species n with b, d, i, j, n; Species o with e, k, o.

Conversely, intense competitive dynamics were observed within blue-clustered regions, suggesting heightened resource and spatial competition. Significant negative correlations were identified between the following species pairs: species a and g/l; species b and g/l; species e and i/n; species f and h; species g and a/b/i; species h and f/j/n; species i and e/g/l/o; species j and h; species l and a/b/d/i; and species o and b/i/j/m.

## Interaction between drivers

We used SEM to analyze the influence of environmental factors on the habitat suitability of 15 invasive mealybug species. The model fit evaluation results show that the overall model exhibits excellent fit. The CFI value is 0.93, the RMSEA is

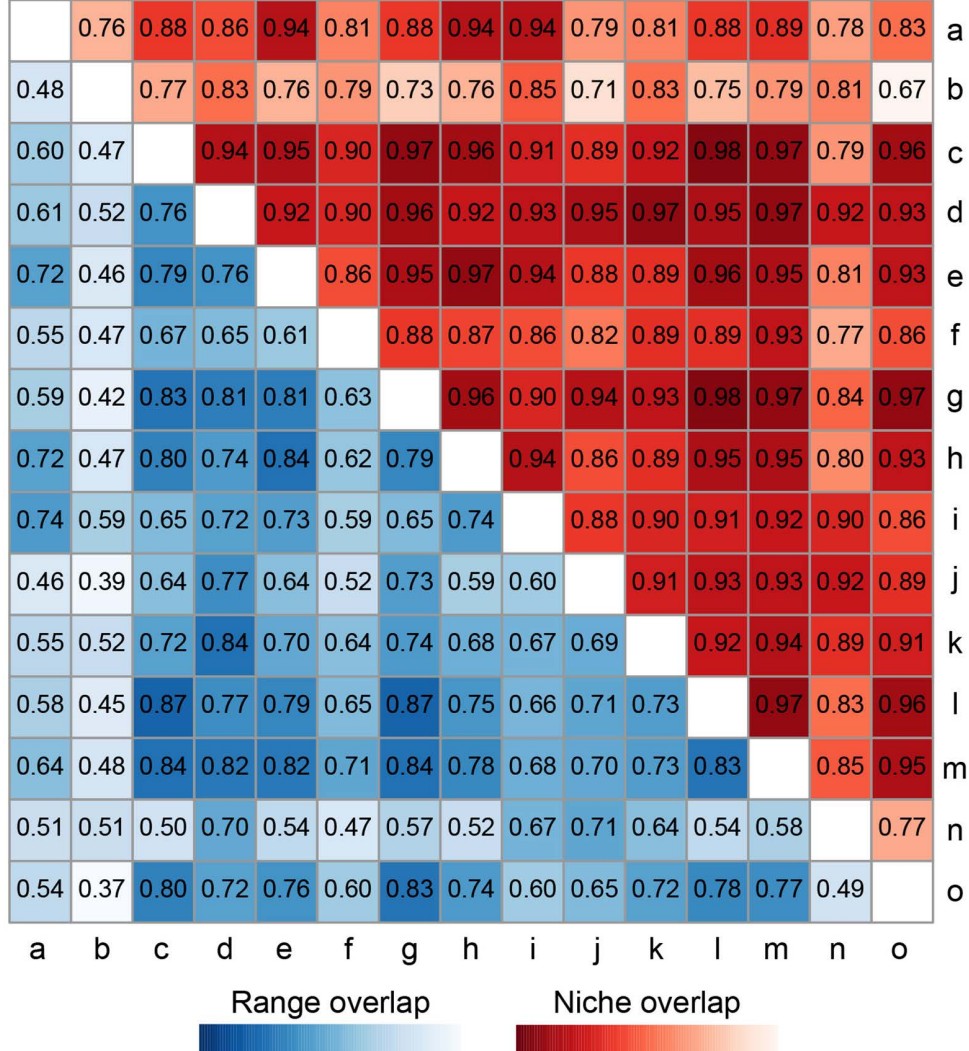

**Fig 2. Niche and range overlap of 15 invasive mealybug (Coccoidea) species in Hainan Island.** a: *Dysmicoccus brevipes*; b: *Icerya purchasi*; c: *Planococcus minor*; d: *Phenacoccus solenopsis*; e: *Pseudococcus jackbeardsleyi*; f: *Phenacoccus madeirensis*; g: *Paracoccus marginatus*; h: *Phenacoccus manihoti*; i: *Phenacoccus parvus*; j: *Planococcus lilacinus*; k: *Aulacaspis yasumatsui*; l: *Ferrisia virgata*; m: *Ceroplastes rusci*; n: *Dysmicoccu neobrevipes*; o: *Parasaissetia nigra*.

0.02, the SRMR is 0.04, and the GFI is 0.97, all of which meet the standards for good SEM fit. These indices indicate that the model effectively captures the underlying relationships in the data, with high explanatory and predictive power. The outstanding performance of these fit indices provides a reliable foundation for further analysis and interpretation of the path coefficients.

Before developing the model, we carried out essential data preprocessing to ensure that the data met the model's assumptions. As depicted in Fig 4, all crucial environmental factors, namely latitude, elevation, climatic conditions, habitat types, and infestation area, exhibited an approximate normal distribution. This characteristic enhanced the reliability of the SEM analysis outcomes.

In subsequent analyses, we explored the relationships between the driving factors and habitat suitability. The results illustrates the linear associations between all environmental factors and the habitat suitability for 15 invasive mealybug

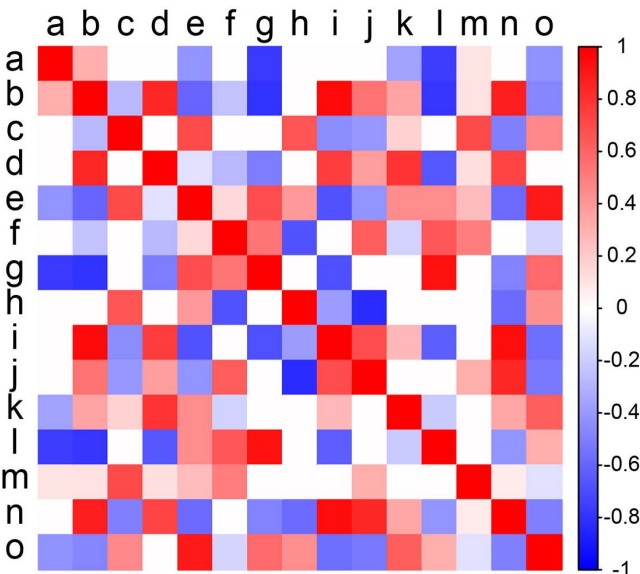

**Fig 3. Species Interaction of of 15 invasive mealybug (Coccoidea) species in Hainan Island.** a: *D. brevipes*; b: *I. purchasi*; c: *P.minor*; d: *P. solenopsis*; e: *P. jackbeardsleyi*; f: *P. madeirensis*; g: *P. marginatus*; h: *P.manihoti*; i: *P. parvus*; j: *P. lilacinus*; k: *A. yasumatsui*; l: *F. virgata*; m: *C. rusci*; n: *D. neobrevipes*; o: *P. nigra*.

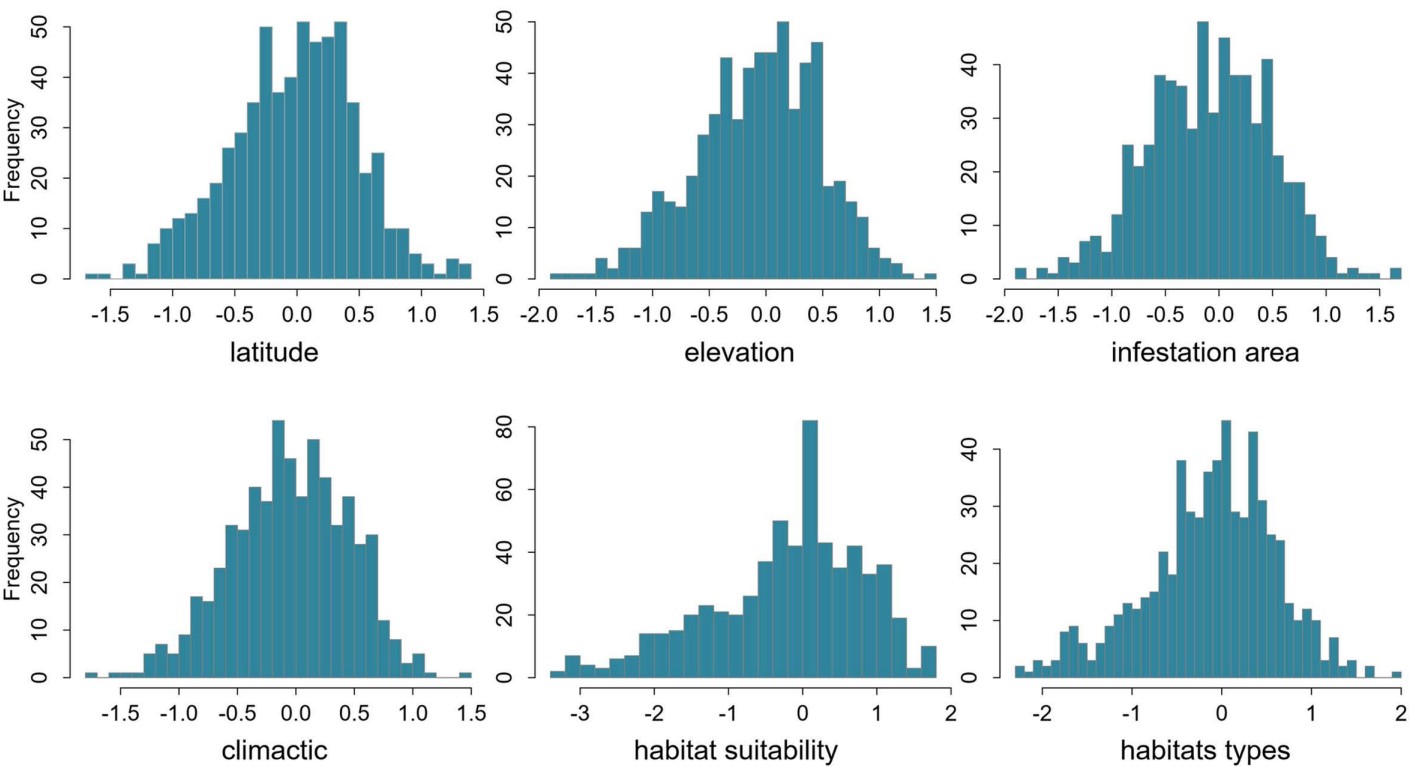

**Fig 4. Normality assessment of environmental predictors (latitude, elevation, climate, habitat types, and infestation area) prior to SEM implementation.**

species (Fig 5). Linear regression analysis showed significant positive relationships between these factors and habitat suitability. Specifically, the regression coefficients and significance levels were as follows: for latitude and habitat suitability, $R^2 = 0.39$, $P < 0.001$; for climatic conditions and habitat suitability, $R^2 = 0.37$, $P < 0.001$; for the infestation area and

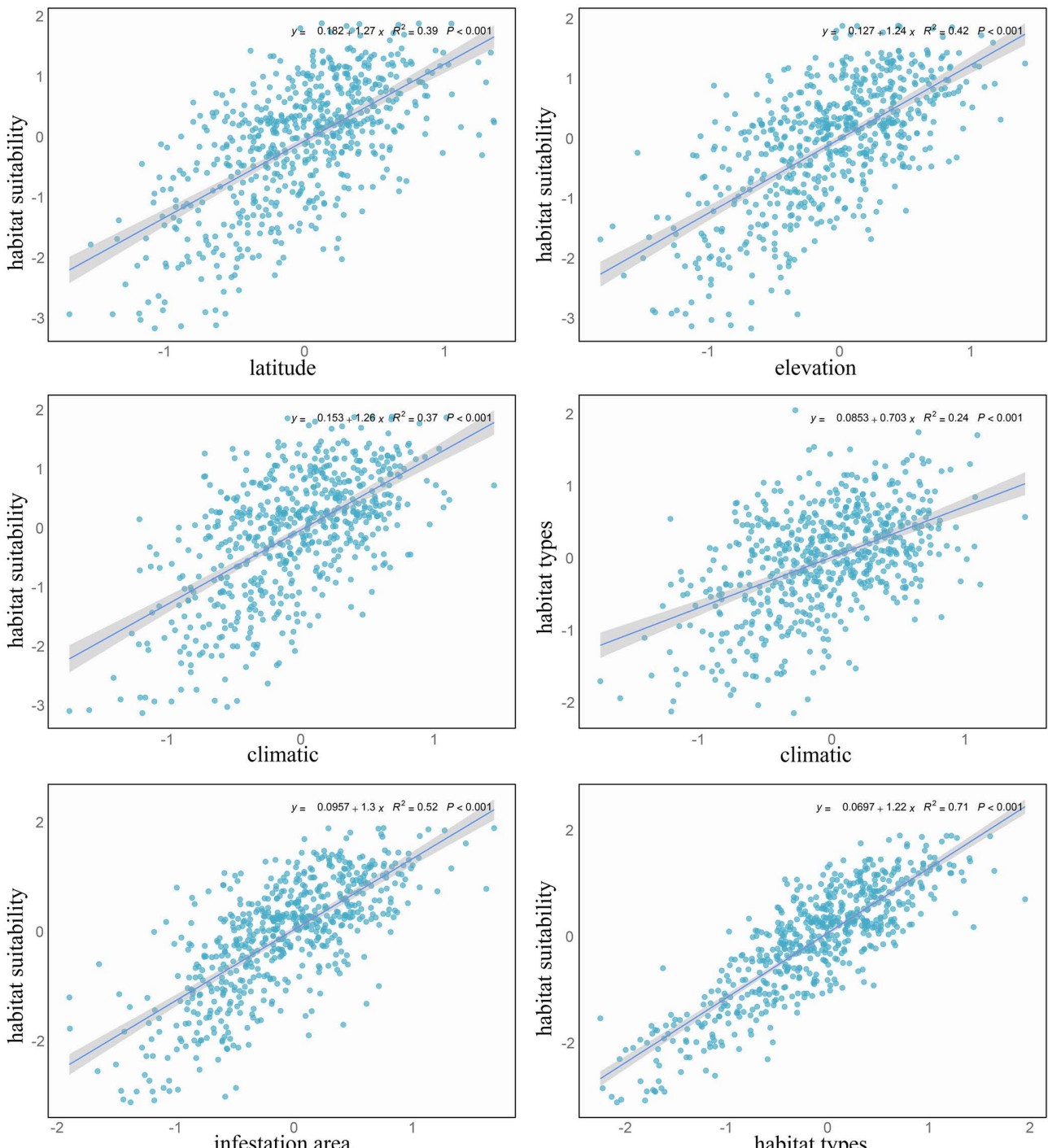

**Fig 5. Linear regression analysis of the relationships between environmental factors.**

habitat suitability, R²=0.52, P<0.001; for the elevation and habitat suitability, R²=0.42, P<0.001; for the climate and habitat types, R²=0.24, P<0.001; and for habitat types and habitat suitability, R²=0.71, P<0.001. Predictor contributions ranked as habitat types > infestation area > elevation > latitude > climatic conditions.

The SEM analysis results indicate that all environmental drivers positively contribute to species habitat suitability, with habitat types exhibiting the strongest comprehensive influence (standardized total effect=0.48) (Fig 6). The total effects of climate, latitude, elevation, and infestation area's area on suitability were 0.20, 0.14, 0.15, and 0.24, respectively, suggesting these factors collectively shape habitat suitability through distinct pathways. Further analysis revealed that climate, latitude, elevation, and infestation area indirectly influenced habitat suitability through their effects on habitat types, with direct path coefficients of 0.15, 0.25, 0.23, and 0.14 on habitat types respectively. This demonstrates their indirect impacts via the "habitat quality-suitability response" cascade effect, where for instance, increasing latitude could indirectly enhance habitat suitability by altering vegetation types.

## Discussion

### Invasive mealybug species richness gradients

The distribution of 15 invasive mealybug species richness in Hainan island follows a pronounced "coastal high-inland low" gradient, with maximum diversity observed in southeastern coastal regions (Haikou, Sanya, Yangpu Port). Three synergistic drivers underlie this spatial pattern: 1) climatic suitability (high temperature/humidity); 2) international trade networks facilitating species introductions via ornamental plants (e.g., hibiscus) and tropical fruits (e.g., pineapple), particularly for high-risk species like *P. solenopsis* and *P. marginatus* [58–59]; and 3) coastal habitat homogenization from mono-culture plantations (rubber, oil palm) that reduce native biodiversity while creating invasion opportunities for generalists (e.g., *P. madeirensis*) through diminished biotic resistance [60–61]. These factors are amplified by invasive ant-mealybug

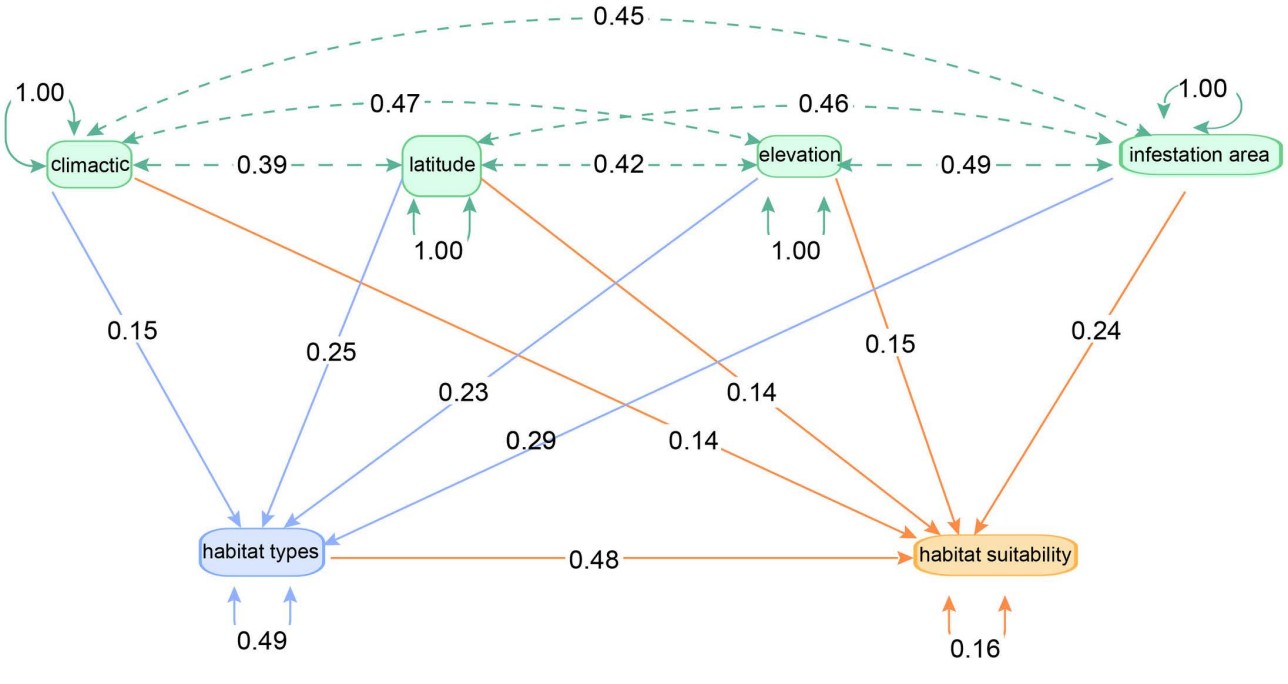

**Fig 6. Effects of drivers based on structural equation modeling (SEM) on habitat suitability for 15 species of mealybugs.**

mutualisms (*S. invicta-Anoplolepis gracilipes* with *P. Nigra*), where ant aggression disrupts natural enemy control (e.g., parasitoid wasp *Metaphycus parasaissetiae*), establishing persistent coastal hotspots through positive feedback [14,62].

Conversely, inland mountainous zones (Wuzhi Mountain, Yingge Ridge) maintain low invasion levels through dual filters: 1) abiotic barriers (topographic complexity, seasonal aridity) and 2) biotic resistance from intact forest ecosystems. Primary forests with multi-layered vegetation sustain diverse predator guilds (lady beetles, lacewings) that exert "natural enemy saturation" effects, effectively suppressing invaders like *I. purchasi* [63]. Seasonal variability further modulates populations: dry season (November-April) ant activity enhances mealybug survival, while monsoon rains (May-October) physically disrupt colonies of key species (*P. minor*, *D. brevipes*) through microhabitat flushing [64–65].

## Niche and range overlap in invasive mealybugs

A niche overlap index of $I \geq 0.67$ among the 15 invasive mealybug species groups highlights significant environmental adaptability convergence in Hainan island. *P. nigra* (o) exemplifies this pattern, showing high niche overlap (>0.9) with most species except *D. brevipes* (a), *F. virgata* (l), and *P. lilacinus* (j). This high overlap is driven by *P. nigra*'s distinctive biological traits: a developmental threshold temperature of 10.74°C, a broad thermal tolerance spanning 12–30°C (wide-temperature), and polyphagy across 160 host species [66–68]. Similarly, other high-overlap species—*P. solenopsis* (d; 55 families, 207 hosts; optimal range 17–32°C), *P. marginatus* (g; 68 families, 264 hosts; 18–30°C), *P.minor* (c; 73 families, >250 hosts; 10.74–32°C), and *P. jackbeardsleyi* (e; 50 families, >200 hosts; 17–32°C with 35°C tolerance)—demonstrate analogous adaptive strategies. These species inflict damage through sap-feeding on leaves and tender shoots by adult females and nymphs, causing leaf wilting, yellowing, deformation, and plant mortality; their concurrent honeydew excretion induces sooty mold proliferation, which impairs photosynthetic capacity and accelerates host decline through synergistic phytopathological effects [58,69,70]. The observed niche overlap in Hainan province arises from convergent environmental adaptation, broad host plasticity, and shared damage mechanisms. This overlap escalates interspecific interference competition—particularly through scramble competition for phloem resources—thereby amplifying population volatility, a pattern mirroring competition-mediated instability documented in aphid assemblages [71–72].

For high-overlap species pairs (Sørensen > 0.8), the extensive co-occurrence of species such as *P.minor* (c) and *P. marginatus* (g) arises not only from their convergent adaptation to tropical climates (24–28°C) but also benefits from their broad host-specific traits [58,64,73]. Furthermore, the range overlap between *P. marginatus* and *F. virgata* (l), *C. rusci* (m), and *P. nigra* (o) highlights their adaptive radiation to microhabitats in economically important crops. Notably, *P. nigra* (o) overcomes niche limitations through its eurythermic (12–30°C) and polyphagous characteristics [62], forming a compound infestation pattern with *P. marginatus* (g), *P. minor* (c), *F. virgata* (l), and *C. rusci* (m) that relies on ant-mediated protection (effectively evading parasitoid wasps and other natural enemies) [68].

In contrast, low-overlap pairs such as *I. purchasi* (b) and *P. nigra* (o) (Sørensen = 0.37) reflect distinct ecological niche differentiation. *I. purchasi*, primarily infesting Rutaceae, Fabaceae, Asteraceae, Rosaceae, and Solanaceae plants [74], exhibits host range constraints that create resource mismatches with *P. nigra*, which prefers rubber trees and exhibits broader thermal adaptability, ultimately leading to low suitability overlap [62]. Additionally, the diversity of habitat types and topographical conditions provides a variety of resources and habitat environments, further promoting species adaptation and coexistence [75].

## Analysis of interspecific interactions

We employed the JSDM to reveal interspecific interactions among 15 invasive mealybug species, uncovering significant synergistic and competitive relationships. In ecology, synergism refers to the phenomenon where species mutually benefit, while competition arises when species vie for shared resources. Studies demonstrate that the significant negative correlations observed in species pairs like *D. brevipes* (a)-*P. marginatus* (g) and *I. purchasi* (b)-*P. marginatus* primarily stem from direct overlaps in ecological requirements: on one hand, shared host strategies (e.g., concentrated damage

on economic crops such as papaya, pomegranate, or pineapple) lead to competition for limited phloem resources; on the other hand, similar microhabitat preferences (e.g., leaf undersides or young shoots) intensify spatial competition, thereby driving exclusion effects [58,65,76]. This competitive relationship is further reinforced by differences in thermal adaptation. For instance, in species pair *P. jackbeardsleyi* (e)-*D. neobrevipes* (n), the niche partitioning between heat-tolerant *P. jackbeardsleyi* (withstands 35°C) and *D. neobrevipes* mealybug (optimal range 8.7–29°C) along thermal gradients, while reducing direct conflict, exacerbates competitive exclusion in overlapping environmental conditions [9,77,78].

In stark contrast to competitive relationships, species pairs like *P. minor* (c)-*P. manihoti* (h), *P. parvus* (e)- *P. marginatus* (g) and g-*P. nigra* (o) exhibit significant synergistic effects. This positive correlation manifests mainly in two aspects: First, host preference differentiation promotes coexistence. For example, the *P. nigra* primarily damages tropical garden plants (*Ficus*, *Hibiscus*) and crops like rubber and sugar apple [61], while *P. marginatus* has a broader host range (264 species across 64 families) including various economic crops like papaya and mango [69]. Similarly, the *P. minor* with an extremely wide host range (264 species across 64 families) covering fruit trees (citrus, mango), ornamental plants (rose, croton) and crops (potato, corn) [58], contrasts with the *P. manihoti* which, despite being oligophagous, focuses 70% of damage on cassava, only temporarily feeding on non-economic hosts like citrus and soybean [79]. Second, reproductive strategy differentiation (e.g., thelytokous parthenogenesis in *P. parvus* with eggs hatching at 25°C and 68% RH versus seasonal outbreaks of *P. marginatus* in spring/autumn) reduces competitive pressure through spatiotemporal niche partitioning [69,78,80].

These interactions collectively constitute the dynamic equilibrium of invasive mealybug communities: antagonistic effects limit local dominance by single species, while mutualistic mechanisms enhance overall invasion resilience [81]. In monoculture systems, initial resource abundance may support simultaneous colonization by multiple species, but subsequent competition filters out species with adaptive advantages like eurythermy and polyphagy (e.g., *P. nigra*) [68]. Niche and range range overlap analyses revealed consistent patterns: species pairs with high overlap predominantly showed competitive relationships, whereas those with low overlap (e.g., *I. purchasi* and *P. nigra*) exhibited facilitation through niche differentiation. JSDM results confirmed these competitive and facilitative interactions, which were further supported by the species' thermophilic preferences, biological traits, and diverse host availability as well as the province's suitable tropical climate and heterogeneous habitat types (a finding corroborated by SEM results).

### Influence of environmental factors on the habitat suitability

SEM revealed that habitat types is the primary driver of habitat suitability for 15 invasive mealybug species, exhibiting both the strongest direct effect (path coefficient = 0.48) and the highest independent explanatory power ($R^2 = 0.71$). The central role of habitat aligns with broader entomological patterns: habitat is a primary driver of global insect population distribution patterns [82–84]. Specifically, our findings resonate with studies on Lepidoptera species such as *Zygaena angelicae* elegans, where habitat fragmentation and microhabitat critically determine population viability [85].

Our results revealed a significant interaction effect between climate and latitude (path coefficient = 0.39), indicating their synergistic regulation of mealybug habitat suitability through microhabitat modification. This aligns with established entomological principles where climatic factors (temperature/humidity) serve as fundamental determinants of insect distributions [86–87], particularly under climate change scenarios [88–90]. The 15 invasive mealybug species studied exhibited characteristic thermo-hygric dependencies: optimal development occurred at 20–30°C (*P. marginatus*: 24–28°C; *P. madeirensis*: 20–30°C), with extreme heat (>35°C) causing significant mortality [64,73,91]. Species-specific humidity adaptations were observed – *D. neobrevipes* showed sensitivity to aridity and heavy precipitation [9], whereas *P. manihoti* demonstrated dry-season proliferation [91], and *P. parvus* required 68% relative humidity for successful oviposition [78–80].

The path coefficients for latitude and elevation on habitat types were 0.25 and 0.23 respectively, indicating that these two geographical factors indirectly determine the suitable range of mealybugs by influencing habitat types. This finding is consistent with multiple research results: insect populations in temperate regions are significantly affected by latitude

[92]; while elevation gradients, by integrating environmental factors such as temperature, and humidity [93–94], markedly alter the insect community structure in mountain ecosystems [87–96]. For example, moth species in Baden-Württemberg, Germany declined by 20–60% between 1970 and 2020 due to elevation changes [85]. Furthermore, the area of infestation was directly related to suitable distribution, with the highest path coefficient (0.24) for suitability. Field surveys confirmed that areas with larger infestation areas indeed exhibited higher habitat suitability.

## Integrated management and niche modeling optimization

To effectively mitigate mealybug invasions, a tripartite strategy should be implemented: Firstly, prioritize control of polyphagous and eurythermic species (e.g., *P. jackbeardsleyi* and *P. solenopsis*) that drive multi-species outbreaks. Apply temperature-triggered biocides during their thermal optimum windows (25–32°C) to maximize efficacy; Secondly, implement species-specific ant baiting to dismantle protective ant-mealybug symbioses (e.g., *S. invicta* with *P. solenopsis* or *A. gracilipes* with *P. nigra*), thereby reducing invasion resilience; Thirdly, establish native vegetation buffers in monoculture plantations to boost natural enemy diversity, particularly parasitoid wasps (Metaphycus spp.) that simultaneously suppress multiple co-occurring mealybugs.

Considering that habitat types and infestation area dominate the studies driving mealybug suitability, it is recommended that ecological niche models be constructed to incorporate them as key predictors into model predictions. This is specified as follows: Firstly, habitat types, as the most dominant direct driver (path coefficient = 0.48), categorize habitats into functional units, such as rubber plantations, orchards, and urban green spaces. Secondly, regarding the infested area, spatiotemporal weighting matrices are incorporated to consider its direct positive effect (path coefficient = 0.24) on habitat suitability, thereby reflecting the invasion feedback dynamics. These recommendations align with ecological theories on climate-driven species distributions [97] and enhance the predictive power of "climate-geography-habitat" multi-scale coupling models for tropical mealybug invasions.

## Limitations of the study and future research directions

This study systematically analyzed the distribution patterns, species richness, niche overlap, and their relationships with environmental factors for 15 invasive mealybug species in Hainan Province, elucidating the mechanisms by which environmental factors influence species suitability. However, certain limitations remain in the application of SEM and JSDM models. Firstly, the limited sample size may have affected the stability of the models and the generalizability of the results, especially in the case of rare species, where the sample collection may not fully represent their distribution patterns [98]. Furthermore, the selection of environmental factors and model assumptions may have constrained the comprehensive capture of complex ecological relationships. For instance, this study did not account for the direct impact of human activities on Coccoidea distribution. Future research could expand the models by incorporating additional variables or adopting multiscale modeling approaches to more accurately capture the relationship between species distribution and environmental factors [99].

The spatial scope of this study is confined to Hainan island, which, while representative, may limit the applicability of the findings to other regions with different geographical environments. Future studies could extend to other areas, particularly tropical and subtropical regions, to test the generalizability of Coccoidea distribution patterns. Additionally, the study's limited temporal scope and reliance on short-term data may not adequately capture climate change's long-term effects on species distribution and population dynamics [100,101]. Consequently, incorporating long-term ecological monitoring data alongside more extensive spatial datasets would enhance the accuracy of species distribution models. Future investigations should prioritize assessing climate change's lasting impacts on Coccoidea distributions, especially through predictive modeling across various climate scenarios. Examining how interspecific interactions vary under different ecological conditions would further elucidate species' adaptive capacities and competitive relationships. Integrating remote sensing data with climate models is also advised to improve large-scale distribution predictions, facilitating more targeted conservation and management strategies for Coccoidea species.

## Conclusion

This study comprehensively analyzed the species richness, niche overlap, range overlap, interspecific interactions, and environmental factors affecting habitat suitability of 15 invasive mealybug species in Hainan Island, revealing their distribution patterns and key ecological drivers. The species richness of invasive mealybugs exhibited a distinct "coastal high–inland low" gradient, with significantly higher richness in southeastern coastal regions (e.g., Haikou, Sanya, Yangpu Port), while inland mountainous areas (e.g., Wuzhi Mountain, Yingge Ridge) showed lower richness due to climatic filtering (complex topography) and strong ecosystem resistance.

Niche overlap analysis indicated significantly elevated niche overlap indices ($I \geq 0.67$) among the 15 invasive mealybug species in Hainan, reflecting environmental adaptability convergence in tropical climates driven by shared thermal tolerance, broad host preferences, and convergent damage mechanisms. Range overlap analysis revealed that high-overlap species pairs (Sørensen > 0.8), such as *P. minor* (c) and *P.marginatus* (g), coexisted widely due to convergent adaptation to humid tropical climates (24–28°C) and trans-family host utilization strategies. In contrast, low-overlap pairs (e.g., *I. purchasi* (b) and *P. nigra* (o), Sørensen = 0.37) exhibited reduced overlap through niche differentiation (e.g., host specificity or microhabitat segregation). Mechanistically, JSDM confirmed these patterns, revealing competitive exclusion where resource overlap and thermal adaptation divergence occurred (e.g., *P. marginatus* vs. *Dysmicoccus brevipes*), versus coexistence promoted by host specialization (e.g., *Ceroplastes psidii*'s rubber tree specificity versus *P. marginatus*'s generalism), reproductive strategy divergence (parthenogenesis vs. seasonal outbreaks), and external environmental factors. Ultimately, SEM analyses and Linear regression identified significant positive correlations between habitat conditions and species suitability ($R^2 = 0.71$, $p < 0.001$), with habitat type as the dominant driver (total effect = 0.48), where climate and topography indirectly regulated suitability through habitat characteristics (e.g., elevation, latitude) while pest-infested areas directly enhanced suitability (path coefficient = 0.24).

However, there are some limitations in this study, such as the constraints in sample size and spatial scope. Future research should expand the study area and time span, incorporating additional environmental variables and large-scale climate data to further validate the species suitability prediction models. Furthermore, a deeper exploration of the dynamic changes in interspecific interactions under different ecological conditions will provide more precise strategies for the management and control of 15 invasive mealybug species.

This study systematically elucidates the distribution patterns and multi-scale drivers of invasive mealybugs in Hainan, thereby addressing critical knowledge gaps regarding ecological processes in tropical multi-species co-invasions. Through quantitative analysis of niche differentiation and interspecific interaction dynamics, we establish a theoretical framework for developing targeted control strategies—particularly for heat-tolerant species management. Importantly, our findings advance the fundamental understanding of community assembly mechanisms during multi-species invasions, while providing actionable insights for integrated pest management in tropical agricultural ecosystems. The novel integration of JSDM-SEM modeling offers methodological innovation by revealing complex interaction networks in biological invasions. Collectively, this work establishes both a scientific foundation and practical guidelines for sustainable agricultural ecosystem management and biodiversity conservation.

## Supporting information

**S1 File. Habitat suitability data for 15 mealybug species.**
(RAR)

**S2 Fig. Species richness distribution of 15 invasive mealybug species.**
(PDF)

**S3 Data. Niche overlap indices.**
(XLSX)

**S4 Data. Range overlap values.**
(XLSX)

**S5 Data. Interaction matrix of 15 invasive mealybugs on Hainan Island.**
(XLSX)

**S6 Data. Source data for** Figs 4–6.
(XLSX)

## Author contributions

**Conceptualization:** Qin Si, Jie Hu, Gaochao Xu.

**Data curation:** Jie Hu, Mulan Ji, yanjing zhang.

**Formal analysis:** Jie Hu, Mulan Ji, Gaochao Xu, yanjing zhang.

**Funding acquisition:** Qin Si, yanjing zhang.

**Investigation:** Zhong Hua.

**Methodology:** yanjing zhang.

**Resources:** Binxin Wu.

**Software:** Jie Hu, Jian Wang, yanjing zhang.

**Supervision:** Jian Wang.

**Visualization:** yanjing zhang.

**Writing – original draft:** Qin Si, yanjing zhang.

**Writing – review & editing:** Qin Si, Binxin Wu, yanjing zhang.

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
