## [Decision Letter · Decision Letter 0]

25 Jul 2025

Dear Dr. zhang,

Thank you for submitting your manuscript to PLOS ONE. After careful consideration, we feel that it has merit but does not fully meet PLOS ONE’s publication criteria as it currently stands. Therefore, we invite you to submit a revised version of the manuscript that addresses the points raised during the review process.

We look forward to receiving your revised manuscript.

Kind regards,

Kleber Del-Claro, PhD

Academic Editor

PLOS ONE

Journal Requirements:

1. Please ensure that your manuscript meets PLOS ONE's style requirements, including those for file naming. The PLOS ONE style templates can be found at https://journals.plos.org/plosone/s/file?id=wjVg/PLOSOne_formatting_sample_main_body.pdf and https://journals.plos.org/plosone/s/file?id=ba62/PLOSOne_formatting_sample_title_authors_affiliations.pdf.

Reviewers' comments:

Reviewer's Responses to Questions

**Comments to the Author**

1. Is the manuscript technically sound, and do the data support the conclusions?

Reviewer #1: Yes

Reviewer #2: Yes

2. Has the statistical analysis been performed appropriately and rigorously?

Reviewer #1: I Don't Know

Reviewer #2: Yes

3. Have the authors made all data underlying the findings in their manuscript fully available?

Reviewer #1: No

Reviewer #2: Yes

4. Is the manuscript presented in an intelligible fashion and written in standard English?

Reviewer #1: Yes

Reviewer #2: Yes

Reviewer #1: Dear authors, I made a few minor suggestions in the text (attached file), and I would like to raise two important points for your consideration.

First, I noticed that some interpretation of the results appears in the “Results” section. I wonder whether this is standard practice in this field to discuss the implications of the findings within the “Results” section itself, since such discussion is typically reserved for the “Discussion” section.

Another issue refers to how field data was obtained. The authors state that “all occurrence data were obtained through systematic surveys,” but they do not provide details on how these surveys were conducted. I would like to see more details in this subject, because the parameters of the survey can affect the outcomes of your study. I assume authors selected the 15 mealybugs species based on this fieldwork, so I would like to see more details of the fieldwork.

A final consideration: the y axis title in figure 5 must be 180 degrees inverted, so the text is aligned like in the y axis of figure 4.

I did not find the data associated with the Analyses.

Reviewer #2: The article is well-written, with all the necessary elements to facilitate the reader's understanding. The way the introduction was written shows a succession of themes and the scope of all the subjects that the text deals with. In addition, they have already shown in the introduction the importance of the methods that were used to evaluate the data. The emphasis on the methodology used is very important and will make this article cited by other researchers, making it an important reading.

The methods have been written down thoroughly, ensuring that they can be replicated. The results corroborate the conclusions at the end of the text and show important data, especially in view of climate change and the anthropogenic effects to which the natural environment is subject. In addition, this work can serve as a basis for decision-making regarding pest control.

It was very important to mention both the shortcomings of the work and the future perspectives so that other researchers can build on this manuscript.

Below, I draw attention to some points of correction:

1. Italicize the word "lavaan" on line 126;

2. Remove the " sign on line 290;

3. Remove a dot on line 469;

4. Remove an extra space on line 481;

5. Talk about the host of the mealybugs in question in this area of study. The authors only inform that it is the coconut tree at the end of the article, in the conclusions of the work.

**Do you want your identity to be public for this peer review?** For information about this choice, including consent withdrawal, please see our Privacy Policy

Reviewer #1: No

Reviewer #2: No

---

## [Author Response · Author response to Decision Letter 1]

31 Jul 2025

Dear Editor and Reviewer:

We extend our sincere gratitude to the Editor for the rigorous review and the opportunity to revise our manuscript. We also deeply appreciate the reviewers' constructive and insightful critiques. Upon receiving the feedback, our research team carefully examined all comments from the Editor, the journal, and the two reviewers, and subsequently implemented comprehensive revisions and refinements to the manuscript, systematically addressing all concerns. The reviewers' perspectives were exceptionally valuable and profound, and their guidance has been instrumental in enhancing the quality of this work. We sincerely acknowledge the considerable time and specialized expertise they dedicated to evaluating our manuscript. Below, we provide detailed point-by-point responses to each comment raised by the Editor and reviewers, along with explanations of the corresponding revisions made. The revised manuscript is now respectfully submitted for your consideration. Should any further refinements be required, we respectfully request the opportunity to undertake additional revisions—our team remains fully committed to perfecting this work until it meets the journal's publication standards. Additionally, prior to the main text, we provide the following clarification regarding author name correction

In the original manuscript, the author name was erroneously presented as "Si Qin" due to inadvertent reversal of surname/given name order. This is strictly a spelling correction, and the name has been restored to the correct form "Qin Si" in the revised manuscript. We explicitly confirm that "Si Qin" and "Qin Si" refer to the same individual author; no change in authorship has occurred. We apologize for any confusion caused by the initial error.

Here is my detailed point-by-point response to the reviewers' suggestions:

Academic editor:

1. Response: Thank you for highlighting this important point. We have supplemented the financial disclosure, as per your request, included it in the updated cover letter.

Response: Thank you for your guidance on revising our manuscript. As requested, we have completed the following actions: specifically, all revised figure files have been successfully uploaded to the Preflight Analysis and Conversion Engine (PACE) platform, with the access path being https://pacev2.apexcovantage.com/Upload# (files are visible under our account). Additionally, we confirm that the figures were processed by PACE to ensure adherence to PLOS ONE’s digital standards.

Response: We appreciate the recommendation to deposit protocols on protocols.io. While we recognize the value of this platform, we have instead provided comprehensive methodological details along with all relevant source data as Supporting Information files, which have now been uploaded to the journal's submission system. These materials include complete experimental procedures, raw datasets, and technical specifications to ensure full reproducibility of our results. Should additional protocol clarification be needed during the review process, we are prepared to provide further information as required.

Reviewer #1:

1. First, I noticed that some interpretation of the results appears in the “Results” section. I wonder whether this is standard practice in this field to discuss the implications of the findings within the “Results” section itself, since such discussion is typically reserved for the “Discussion” section.—Characterized by dense forest vegetation, high elevation, rich native biodiversity, and relatively limited human disturbance, these zones demonstrate notable resistance to invasive species establishment.

Response: Thank you for highlighting this important point. We agree that interpreting results within the "Results" section is not standard practice, as such discussion should be reserved for the "Discussion" section. The sentence in question has been deleted without delay, and we have ensured that the "Results" section now strictly presents objective findings. We appreciate your meticulous review, which significantly improved the manuscript's rigor, as detailed in lines 261-263 of the Revised Manuscript.

2. Another issue refers to how field data was obtained. The authors state that “all occurrence data were obtained through systematic surveys,” but they do not provide details on how these surveys were conducted. I would like to see more details in this subject, because the parameters of the survey can affect the outcomes of your study. I assume authors selected the 15 mealybugs species based on this fieldwork, so I would like to see more details of the fieldwork.

Response: Thank you for your constructive feedback. We have supplemented this section as you suggested, as detailed in lines 141-146 of the Revised Manuscript.

3. A final consideration: the y axis title in figure 5 must be 180 degrees inverted, so the text is aligned like in the y axis of figure 4.

Response: We have corrected Fig 5 as you suggested, see Fig 5 in the revised version for more details.

4. I did not find the data associated with the Analyses.

Response: We have supplemented additional data to address this concern. The complete datasets are provided in the Supporting Information (uploaded as a compressed file to the journal’s submission system for your review).

5. Attachment: Response to Reviewer Comments for Manuscript (Rev PONE-D-25-29226)

1) Comment on Line56: "insert space"

Response: This issue has been corrected. Please see Line 51 in the revised manuscript.

2) Comment on Line 64: "Scientific names in italics"

Response: This issue has been corrected. Please see Line 63 in the revised manuscript.

3) Comment on Line 69: "Italics"

Response: This issue has been corrected. Please see Line 69 in the revised manuscript.

4) Comment on Line142-144: "This deserves more explanation. How was the survey actually conduted? How many fieldtrips were made, which was the richness and diversity of speceis, how many plots were surveyed, and etc. I believe this survey can justify the study of the 15 mealybug species, right?"

Response: Thank you for your constructive feedback. We have supplemented this section as you suggested, as detailed in lines 141-146 of the Revised Manuscript.

5) Comment on Line 261: "insert space"

Response: This issue has been corrected. Please see Line 263 in the revised manuscript.

6) Comment on Line 261-263: "Characterized by dense forest vegetation, high elevation, rich native biodiversity, and relatively limited human disturbance, these zones demonstrate notable resistance to invasive species establishment."

Response: This sentence was removed as you suggested. Please see Line 261-263 in the revised manuscript.

7) Comment on Line 270: "The total Hainan area is not equal to 100% (18.5 + 37.33 + 44.13 = 99.96% or the area)."

Response: We have verified the data and corrected it to 18.54%, with all values now uniformly reported to two decimal places in the revised manuscript. Please see Line 259 in the revised manuscript.

8) Comment on Line348-349行�"correlation is not a regression"

Response: Thank you for your precise feedback. We acknowledge the inappropriate use of "correlation" when reporting regression results. The description has been revised to clarify that these are R² values from linear regression models (not correlation coefficients). Please see Line 346-347 in the revised manuscript.

9) Comment on Line 389: "purchasi"

Response: This issue has been corrected. Please see Line 386 in the revised manuscript.

10) Comment on Line 392: "insert space"

Response: This issue has been corrected. Please see Line 389 in the revised manuscript.

11) Comment on Line 392: "brevipes"

Response: This issue has been corrected. Please see Line 389 in the revised manuscript.

12) Comment on Line 420: "insert space"

Response: This issue has been corrected. Please see Line 417 in the revised manuscript.

13) Comment on Line 433: "insert space"

Response: This issue has been corrected. Please see Line 430 in the revised manuscript.

14) Comment on Line 434: "insert space"

Response: This issue has been corrected. Please see Line 431 in the revised manuscript.

15) Comment on Line 449: "italics"

Response: This issue has been corrected. Please see Line 446 in the revised manuscript.

Reviewer #2:

1. The article is well-written, with all the necessary elements to facilitate the reader's understanding. The way the introduction was written shows a succession of themes and the scope of all the subjects that the text deals with. In addition, they have already shown in the introduction the importance of the methods that were used to evaluate the data. The emphasis on the methodology used is very important and will make this article cited by other researchers, making it an important reading.

The methods have been written down thoroughly, ensuring that they can be replicated. The results corroborate the conclusions at the end of the text and show important data, especially in view of climate change and the anthropogenic effects to which the natural environment is subject. In addition, this work can serve as a basis for decision-making regarding pest control.

It was very important to mention both the shortcomings of the work and the future perspectives so that other researchers can build on this manuscript.

Response: We sincerely appreciate your insightful evaluation of our manuscript. Your affirmation of the work’s scientific rigor and applied significance is profoundly encouraging. We are honored by your prediction that this article "will be cited by other researchers," and we pledge to continue advancing this critical field.

Below, I draw attention to some points of correction:

1. Italicize the word "lavaan" on line 126;

Response: This issue has been corrected. Please see Line 116 in the revised manuscript.

2. Remove the " sign on line 290;

Response: This issue has been corrected. Please see Line 290 in the revised manuscript.

3. Remove a dot on line 469;

Response: This issue has been corrected. Please see Line 466 in the revised manuscript.

4. Remove an extra space on line 481;

Response: This issue has been corrected. Please see Line 478 in the revised manuscript.

5. Talk about the host of the mealybugs in question in this area of study. The authors only inform that it is the coconut tree at the end of the article, in the conclusions of the work.

Response: We have addressed this point by adding the host plant information (coconut palms) to the Introduction section. Please see line 57-61 of the revised manuscript for details. The mention in the Conclusions has been removed as suggested. Please see line 548-550 of the revised manuscript for details.

---

## [Decision Letter · Decision Letter 1]

18 Sep 2025

Mechanisms of Multi-Species Mealybug Invasions in Hainan Island of China: Integrating Niche, Distribution, and Habitat Drivers

PONE-D-25-29226R1

Dear Dr. Yanjing Zhang,

We’re pleased to inform you that your manuscript has been judged scientifically suitable for publication and will be formally accepted for publication once it meets all outstanding technical requirements.

Kind regards,

Kleber Del-Claro, PhD

Academic Editor

PLOS ONE

Additional Editor Comments (optional):

Please, consider all the suggestions of both reviewers and answer each one in datails, possible we will avoid an additional round.

Reviewer #1:accept

Reviewer #2: accept

Reviewers' comments:

Reviewer's Responses to Questions

**Comments to the Author**

Reviewer #1: All comments have been addressed

Reviewer #2: All comments have been addressed

2. Is the manuscript technically sound, and do the data support the conclusions?

Reviewer #1: Yes

Reviewer #2: Yes

3. Has the statistical analysis been performed appropriately and rigorously?

Reviewer #1: Yes

Reviewer #2: Yes

4. Have the authors made all data underlying the findings in their manuscript fully available?

Reviewer #1: Yes

Reviewer #2: Yes

5. Is the manuscript presented in an intelligible fashion and written in standard English?

Reviewer #1: Yes

Reviewer #2: Yes

Reviewer #1: Dear authors, I see that the suggestions were addressed and in fact the MS has more quality than the previous versions. I have no further comments.

Reviewer #2: (No Response)

**Do you want your identity to be public for this peer review?** For information about this choice, including consent withdrawal, please see our Privacy Policy

Reviewer #1: No

Reviewer #2: No

---

## [Editor Report · Acceptance letter]

PONE-D-25-29226R1

PLOS ONE

Dear Dr. zhang,

I'm pleased to inform you that your manuscript has been deemed suitable for publication in PLOS ONE. Congratulations! Your manuscript is now being handed over to our production team.

Kind regards,

on behalf of

Dr. Kleber Del-Claro

Academic Editor

PLOS ONE